# Positive Effects on Emotional Stress and Sleep Quality of Forest Healing Program for Exhausted Medical Workers during the COVID-19 Outbreak

**DOI:** 10.3390/ijerph19053130

**Published:** 2022-03-07

**Authors:** Yunsoo Kim, Yoonhee Choi, Hyeyun Kim

**Affiliations:** 1Department of Nursing, College of Medicine, Catholic Kwandong University, Gangneung 25601, Korea; agneskim4320@gmail.com; 2The Convergence Institute of Healthcare and Medical Science, College of Medicine, Catholic Kwandong University, Incheon 22711, Korea; uni2442@gmail.com; 3Department of Neurology, College of Medicine, Catholic Kwandong University, Incheon 22711, Korea

**Keywords:** COVID-19, sleep, stress, medical workers, forest, healing

## Abstract

This study targeted medical workers, who are currently being subjected to an excessive workload and emotional stress during the COVID-19 outbreak. Various treatment programs, such as a relaxation program to relieve stress, a walk in the forest, and woodworking were provided to the participants as forest healing therapies. We enrolled 13 medical workers (11 females, 2 males). Before and after forest healing therapy, stress and sleep-related questionnaires and levels of salivary cortisol, dehydroepiandrosterone sulfate (DHEA-S), and melatonin were measured and compared. The improvement of the perceived stress scale and the decrease of DHEA-S, a stress index, showed statistically significant results. However, although this study was conducted with a small number of participants and has a limitation in that the therapy occurred over a short period of only 1 night and 2 days, the trend of supporting results remains positive. As such, the authors propose forest healing therapy as one intervention to relieve the job stress for this group of workers

## 1. Introduction

Due to the global COVID-19 pandemic, workers are increasingly exhausted, and negative emotional deterioration is becoming more frequent due to social isolation, such that a new term ‘Corona Blue’ has appeared [1]. In the workplace, both a decrease in emotional intelligence and a decrease in work performance have correspondingly been reported. Job stress is a now new topic in the Corona Blue era, and it has become a major problem that can threaten the health of workers [2]. Common adverse symptoms of stress in individuals may include mental health effects such as mood changes, sleep disorders, oral/gum diseases, skin diseases, headaches, and back pain. In terms of productivity, a decrease in employee morale, an increase in opportunity costs, and decreased employee performance could also be considered negative factors [3].

Forest Bathing (Shinrin-Yoku), forest walking, and forest therapy programs refer to programs that allow somebody to spend time in the forest and pursue healing. As many studies have been accumulated on the mental and physiological effects of this since the 1990s, forest healing has attracted attention as a way of healing therapy [4]. Forest bathing positively affects physical health [5], such as strengthening human immunity [6] and treating chronic diseases [7]. Forest bathing also positively affects mental health, with reports stating improvements in mood regulation, anxiety, and depression [8,9]. 

Psychologically high-risk groups include medical workers, students, and public officials, and as such psychological preventions in the COVID-19 outbreak are regarded as a major issue [10]. Recent studies have shown that the physical and mental stress of medical workers during the COVID-19 outbreak is a serious problem [10,11]. In addition, it was reported that the sleep and mental health of frontline medical workers during the outbreak in China was extremely poor [11,12]. In addition, sleep disturbance was found to be more severe in women and required an urgent intervention [13]. The increase in mental exhaustion and sleep disturbance among medical workers due to the COVID-19 outbreak has also been mentioned as an important concern in the press [12]. However, though there has been no lack of awareness of the mental health issue and the need for an urgent psychological intervention, the intervention plan for this psychological crisis remains insufficient.

The emotional crisis of medical workers may be deemed to be less significant for regular citizens compared to the more easily seen issues such as economic problems, social isolation, the emergence of coronavirus variants, and vaccinations related to the COVID-19 outbreak. However, just because the emotional exhaustion of medical workers does not appear to be a real problem, it should not be regarded as a less serious social problem. This study was designed to confirm the positive effects of forest healing on the stress and sleep issues of medical workers, those who are frequently regarded as having markedly increased job stress during the COVID-19 outbreak.

## 2. Materials and Methods

### 2.1. Participants and Study Design

The final enrolled participants for the forest healing program included 13 medical workers: 11 women and 2 men. This study was conducted for those who were working in medical institutions from 2019 to September 2021, during the initial COVID-19 outbreak. Participants were recruited through an announcement to recruit those who increased job loading and became more stressed due to the COVID-19 outbreak. After reviewing the purpose of the study and providing consent to participate, participants entered a forest healing program for two days and one night, in which a self-report questionnaire was completed before and after the program. Physiological parameters including salivary cortisol [14], melatonin [15], dehydroepiandrosterone sulfate (DHEA-S) [16] were also measured. Exclusion criteria included: (1) people with serious diseases such as cancer, (2) people with physical problems that would make it difficult to participate in the forest healing program, (3) people with allergies to certain plants, due to the inherent nature of participation in the forest program, (4) those with depression, anxiety, and severe insomnia that would make it difficult to carry out daily life during the screening interview, and (5) those with a history of heart and cerebral diseases. 

### 2.2. Design of the Forest Therapy Program

The forest therapy program was conducted in November 2021. According to the purpose of this study, since high-stress medical workers were targeted, the forest healing program was primarily comprised of programs that can reduce their stress. The program was developed by applying forest healing factors such as landscape, sound, and sunlight, and forest healing therapies such as plant therapy, exercise therapy, and climate therapy. The overall program used in this study is shown in Table 1 and Figure 1. These programs were conducted at the National Center for Forest Activities in Hoengseong, Gangwon-do, Korea, a forest education and healing center. The Center is located 680 m above sea level. This location features six trails ranging from 450 m to 2000 m in length, and trees such as larch, birch, dogwood, and pine trees naturally grow there. The region has an average temperature of 5.5 °C and a maximum temperature of 11.2 °C in the month.

### 2.3. Evaluation of Sleep and Emotional Status, Laboratory Measurements

Participants conducted assessments of emotion, sleep state, somatization using the Epworth sleepiness scale (ESS) [17], the Stanford sleepiness scale (SSS) (to evaluate the daytime somnolence) [18], the Korean version of the Pittsburgh Sleep Questionnaire Index (PSQI) [19] to investigate overall sleep quality, and the insomnia severity index to measure the severity of insomnia symptoms, the Hospital Anxiety and Depression Scale (HADS) [20], the perceived stress scale (PSS) [21], and somatization symptoms (KSCL95; Korean-Symptom Checklist 95) [22] before entering and after completing the forest therapy program. 

To obtain objective data pertaining to the stress and sleep status, saliva tests were performed twice (4 tests a day) before and after the forest healing program. For the cortisol test, saliva was collected immediately after waking up in the morning, and in order to measure melatonin, saliva was collected immediately before going to bed. The sampling time was 8:00 a.m., after waking up in the morning, and the sleeping collection time was at 9:00 p.m.

This study was conducted following the Declaration of Helsinki, and the protocols were approved by the appropriate ethics review board (#IS21ONSE0077). All participants gave their consent prior to participating in the study.

### 2.4. Statistical Analysis

The collected data were analyzed using SPSS 22.0 (SPSS for Windows, SPSS Inc., Chicago, IL, USA) and AMOS version 22.0 (IBM Corp., Amonk, NY, USA). Specifically, as descriptive statistics, variables related to participants’ general characteristics were analyzed in terms of frequency, percentage, mean, and standard deviation. Changes in the psychological and physiological variables related to stress and sleep were analyzed using a Wilcoxon signed-rank test, box-and-whisker plots, and scattered plots.

## 3. Results 

### 3.1. General Characteristics of Participants

Thirteen participants were enrolled. Among 13 participants, 11 were daytime workers, and 2 were working in three shifts. Daytime workers worked from 9 a.m. to 6 p.m. Although working hours did not increase due to the COVID-19 outbreak, they complained of an increase in work intensity that they felt psychologically. The general characteristics of the study participants are shown in Table 2. The average age of the participants was 42.2 ± 10.99 (mean ± SD) years, with 11 females and 2 males. 

### 3.2. Changes of the Psychological Variables of the Participants

Table 3 shows the results of changes in the psychological variables of the participants before and after the forest therapy. From the study, the participants’ sleep onset time, sleep duration, PSQI, ISI, SSS, HADS, PSS, and somatization were seen to decrease after the forest treatment, with the PSS decrease being statistically significantly; from 30.92 points before forest therapy to 28.23 points after therapy, and sleep duration increased from 332.31 min before forest therapy to 373.85 min after therapy.

Although not statistically significant, the forest treatment was deemed to be effective in relieving insomnia and daytime sleepiness, and in relieving depression, anxiety, and somatic symptoms. 

### 3.3. Changes of the Physiological Variables of the Participants

Table 4 and Figure 2 show the results of changes in the psychological variables of participants before and after the forest treatment program. Participants’ cortisol levels increased before and after forest therapy, whereas the melatonin and DHEA-S levels decreased. DHEA-S decreased statistically significantly from an average of 5.09 before forest treatment to an average of 3.67 after forest treatment.

Melatonin was found to decrease, on average, though it was found to be secreted in 4 out of 13 patients after not being secreted initially. This finding is thought to be related to the ISI and SSS lowering due to changes in the psychological variables of the participants.

The ratio of DHEA-S and cortisol showed a tendency to decrease, from an average of 20.92 before forest therapy to 14.91 after forest therapy, though there was no statistical significance displayed.

The Appendix A show a regression model with interaction showing the relationship between DHEA-S and sleep duration and the relationship between PSS and sleep duration before and after forest healing therapy. The model can be written as follows: before forest healing therapy, estimated 1n (DHEA-s) = 13.79 + 0.06 × sleep duration, after forest healing therapy, estimated 1n (DHEA-s) = 3.48 + 0.02 ×sleep duration. And before forest healing therapy, estimated 1n (PSS) = 20.32 + 0.03 × sleep duration, after forest healing therapy, estimated 1n (PSS) = 21.03 + 0.02 × sleep duration.

### 3.4. Satisfaction Survey

Table 5 shows the participant’s satisfaction of the forest healing program. From the forest healing program satisfaction survey, the average value of overall satisfaction was 4.23, and the mean of each forest healing program was 4.15 to 4.54, indicating a relatively high level of satisfaction among participants. Specifically, satisfaction with ‘dry foot bath (4.54 ± 0.660)’ was the highest, and ‘outdoor activities (4.46 ± 0.877)’, ‘drinking tea-conversation (4.46 ± 0.776)’, ‘woodworking (4.46 ± 0.519)’, ‘warm-up exercises (4.46 ± 0.519)’, and ‘walking (4.46 ± 0.660)’ all showed the second-highest satisfaction, with ‘singing bowl meditation (4.15 ± 0.801)’ recording the lowest satisfaction. In addition, the intention to re-participate in each forest healing program was higher in the order of ‘dry foot bath (4.46 ± 0.660)’, ‘drinking tea-conversation (4.38 ± 0.768)’, ‘walking (4.31 ± 0.751)’, ‘outdoor activities (4.23 ± 0.927)’, ‘wood working (4.00 ± 0.707)’, ‘warm-up exercises (4.00 ± 0.707)’, and ‘singing bowl meditation (3.92 ± 0.954)’. In the case of intention to recommend to others, ‘dry foot bath (4.38 ± 0.650)’ was the highest, followed by ‘drinking tea-conversation (4.38 ± 0.650)’, ‘walking (4.38 ± 0.768)’, ‘outdoor activities (4.31 ± 0.947)’, ‘warm-up exercises (4.23 ± 0.725), ‘woodworking (4.15 ± 0.801)’ and ‘singing bowl meditation (4.15 ± 0.801). Considering these results, in terms of satisfaction, intention of re-participation, and recommendation to others, it could be confirmed that the participants consistently positively evaluated ‘dry foot bath’, ‘drinking tea-conversation’, and ‘walking’ programs. Therefore, it seems that it is necessary to consider these programs when developing a forest healing program for high-stress risk groups.

## 4. Discussion

From this study, the sleep enhancement and emotional stress reduction effects of forest healing were verified, even in a short schedule of 2 days and 1 night. To our knowledge, there has been no research data suggesting an appropriate duration of forest healing therapy in previous studies. In our previous studies, used to evaluate the effectiveness of forest therapy for patients with gastrointestinal malignant diseases and postmenopausal insomnia patients, the studies were conducted for 5 nights and 6 days [23,24]. There has been no report as to the most appropriate duration of forest healing therapy for achieving a proper effect on healthy participants with no specific medical disorders. Participants have the same time to end the same program and return to their room. We checked the bedtime the day after the program, and all the participants answered that they fell asleep at a time similar to their usual bedtime. The effect of improving sleep through a single day of forest healing seems to be more related to the improvement of sleep quality than the change in sleep time or the increase in sleep duration. Despite the limitations of this point, based on the results of this study conducted on healthy workers with job stress, the period for positive effects on sleep and stress is suggested to be 1 night and 2 days.

In this study, the mean level of salivary melatonin increased after forest healing therapy, though with no statistical significance. Interestingly, in 4 out of 13 participants in this study, the level of salivary melatonin was very low that it could not be measured in the laboratory test prior to the forest healing program. After therapy, the level of salivary melatonin increased, with various levels revealed. Since there have been no studies on the relationship between forest therapy and salivary melatonin, further studies are needed in order to determine its relationship and mechanisms of forest healing therapy. From the results of this study, the mechanism of sleep enhancement by which forest healing therapy increased melatonin secretion and consequently improved sleep quality could be considered. More research is also needed in order to explain the mechanism by which forest healing therapy induces melatonin secretion. However, with a review of prior studies, the relationship between melatonin and autonomic nervous system stability, the recovery of disrupted circadian rhythms, and an induction of mood stabilization, are likely to be induced by forest healing therapy, and as a result, it is possible to suggest a hypothesis pertaining to the increased secretion of melatonin [25,26,27,28].

DHEA-S is a commonly studied stress marker [16,29], and the level in this study showed a significant decrease after the forest healing therapy. This reduction was the result of the objective stress improvement due to the forest healing therapy. We present changes according to DHEA-S and sleep duration after forest healing as Appendix A. Due to the small number of participants, the evidence for the equation presented is not strong. Nevertheless, this result is meaningful in that the DHEA-S level after forest healing therapy reduces the effect on sleep duration. In other words, the forest healing program reduces the change in the degree of stress according to the change in sleep duration. Clinically interpreted, it can be suggested that the degree of stress can be lowered even if the sleep duration is the same as usual or a little short when performing the forest healing program. The DHEA-S/cortisol ratio is an objective indicator of the subject’s stress level and may vary depending on gender and age [16,30]. Although there was no statistical significance in the participants’ DHEA-S/cortisol ratio, it tended to decrease after forest treatment, compared to the ratio prior to treatment. Due to the small-sized study, it is deemed necessary to allocate groups by further considering gender and age for future research and to perform additional analyses.

In this study, the results from the forest healing therapy revealed positive effects in terms of relieving insomnia and daytime sleepiness, as well as in relieving depression, anxiety, and somatic symptoms, based on the results of a questionnaire. However, there remains no statistical significance due to the small size of the participants. Further studies with a larger sample size are needed in order to evaluate the psychological positive effects of a short-duration forest healing therapy program. 

This study has several limitations. First, although medical workers were targeted, only 11 nurses and 2 clinical pathologists participated in this program. The limitation is whether they represent medical workers, as only some occupational groups are included. Second, this study only included two shift workers. Shift working is a common type of medical work. Another limitation of this study was that medical workers with various types of work did not participate. Third, salivary tests for melatonin/cortisol were performed, but the objective indicators for stress and sleep were deemed to be insufficient. A further study based on objective variables for sleep/stress measurement is needed. Fourth, the duration of the program was short. Usual forest staying programs range from 3 to 14 nights. Our study was a bit short due to the inherent limitations placed on gathering large numbers of people, due to the COVID-19 pandemic. Fifth, the seasonal effects of forest healing programs were excluded. This study was conducted in early winter, and the cold weather had several limitations on the possible types of physical activity. Sixth, the sample size was small. As this study is a pilot study, the statistical significance of psychological factors and sleep could not be confirmed due to the small size of participants. A large-scale study is thus required in order to analyze the detailed effects of stress and sleep quality improvement and will be extended by conducting research over various periods and seasons, using a more varied sample of medical workers.

Despite these limitations, this study was conducted under the difficult situation of the COVID-19 outbreak. This was the first clinical trial for improving the stress and sleep quality of a one-night forest healing program targeting medical workers. At this time, when social issues about the accumulation of medical worker stress in the COVID-19 outbreak are emerging, our study is a meaningful report as one source of healing for them.

## 5. Conclusions

This study was the first to study the clinical effect of forest healing therapy for medical workers who are currently under a high degree of job stress during the COVID-19 outbreak. Although it has been reported that medical workers’ excessive workload and emotional stress have increased and that there is a need for appropriate intervention was recognized, no appropriate intervention has been suggested [31,32,33,34,35]. The authors have reviewed several previous papers that reported that a forest healing program has a positive healing effect on the emotional stress and bedtime sleep conditions of participants [23,24,36]. These results suggested that forest healing therapy could be helpful in enhancing the emotional healing and positive effects on the sleep of medical workers, who are increasingly at risk of miscellaneous stress during the COVID-19 outbreak. There is still no promise of an end to COVID-19. At the beginning of the COVID-19 outbreak, the main sources of stress for hospital workers included exposure to an infectious agent, stress from exposure to an unspecified group with a high potential for infection, and stress from wearing protective equipment to protect against infection. However, the fact that nobody knows when COVID-19 will end is even more stressful, such that the number of medical workers at risk of stress due to the COVID-19 outbreak is increasing. Now is the time to actively seek out and implement interventions that will relieve their stress. The forest healing program has the potential to give them true rest, enabling them to forget the stress of the COVID-19 outbreak for a while. These authors propose forest healing therapy as one of the intervention methods.

## Figures and Tables

**Figure 1 ijerph-19-03130-f001:**
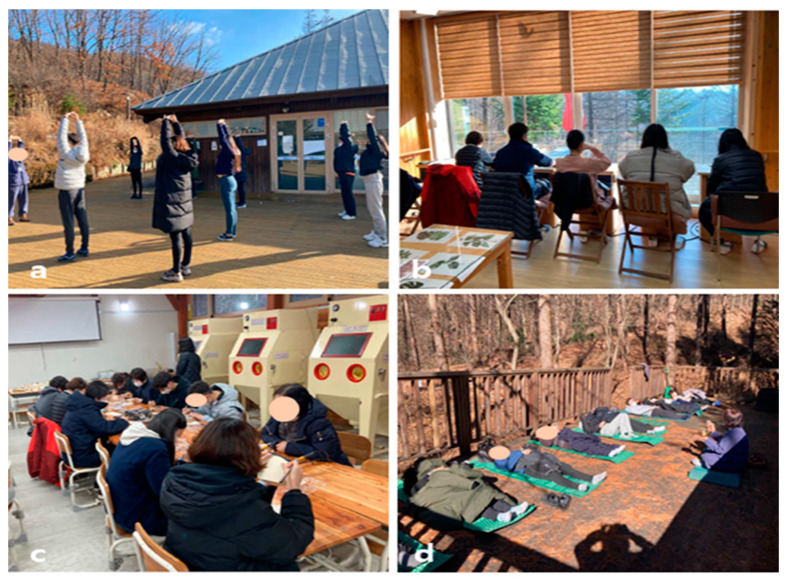
Participants performing activities such as: (**a**) warm-up exercises early in the morning, (**b**) light exposure with tea time at 11:00 a.m., (**c**) making crafts with wood, and (**d**) singing bowl meditation in the evening.

**Figure 2 ijerph-19-03130-f002:**
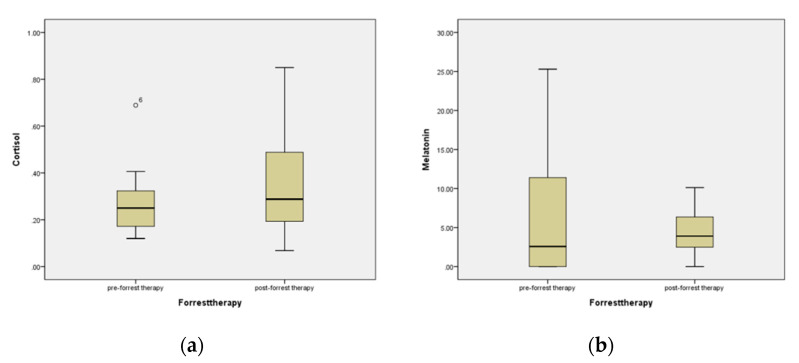
Changes in the level of (**a**) cortisol, (**b**) melatonin, (**c**) DHEA-S, and (**d**) DHEA-s/Cortisol ratio in participants before and after forest therapy.

**Table 1 ijerph-19-03130-t001:** Descriptions of the Forest Healing Program elements.

Program	Contents
Outdoor activities	Activities that awaken the five senses by immersion in elements of the forest, such as sound, scenery, and air while walking on a forest path
Dry foot bath	Activities that use a dry cypress foot bath to warm the feet to relieve foot fatigue and enhance blood circulation
Drinking tea conversation	Activities focusing on the enjoyment of the color, smell, and taste of tea while drinking tea, and talking with the participants in a comfortable atmosphere
Woodworking	Activities promoting a sense of immersion and accomplishment while using a burning machine to engrave favorite sentences, pictures, etc., onto wood carvings
Warm-up exercises	Activities that relieve tension by slowly relaxing the muscles of the entire body
Walking	Activities that give rest to the mind while taking a walk in the forest
Singing bowl meditation	Activities used to empty the head and calm the mind and body while meditating using a singing bowl

**Table 2 ijerph-19-03130-t002:** General characteristics of the participants (N = 13).

	N (%)/Mean ± SD	Min	Max
Age (year)	42.23 ± 10.99	23.00	58.00
Sex (female/male)	11/2 (84.6/15.4)		
Height (cm)	160.5 ± 5.3	152.0	171.0
Weight (kg)	56.8 ± 6.1	50.0	70.5
BMI (cm/kg^2^)	22.1 ± 2.2	19.7	26.0

**Table 3 ijerph-19-03130-t003:** Changes of the psychological variables of the participants (N = 13).

	Pre	Post	Diff (Post-Pre)	Z	*p*
Sleep onset time (min)	33.46 ± 19.94	42.69 ± 38.00	9.23 ± 26.05	1.06	0.291
Sleep duration (min)	332.31 ± 56.74	373.85 ± 61.99	41.54 ± 46.70	−2.54	0.011
PSQI-K	8.00 ± 2.38	6.85 ± 2.61	−1.15 ± 2.15	−1.72	0.086
ISI	8.62 ± 3.01	8.54 ± 4.72	−0.08 ± 3.73	−0.102	0.918
SSS	2.62 ± 1.50	2.08 ± 0.49	−0.54 ± 1.27	−1.51	0.131
ESS	7.38 ± 3.48	8.46 ± 3.57	1.08 ± 3.01	−0.99	0.324
HADS	12.08 ± 6.01	9.69 ± 6.56	−2.38 ± 4.19	−1.84	0.066
PSS	30.92 ± 4.19	28.23 ± 5.00	−2.69 ± 3.73	−2.25	0.025
Somatization symptoms (KSCL95)	14.23 ± 7.67	13.15 ± 9.06	−1.08 ± 6.45	−0.60	0.552

PSQI: Pittsburgh Sleep Quality Index-Korean, ISI: Insomnia Severity Index, SSS: Stanford Sleepiness Scale, ESS: Epworth Sleepiness Scale, HADS: Hospital Anxiety and Depression Scale, PSS: Perceived Stress Scale, KSCL95: Korean-Symptom Checklist 95.

**Table 4 ijerph-19-03130-t004:** Changes of the physiological variables of the participants (N = 13).

	Pre	Post	Diff (Post-Pre)	Z	*p*
Cortisol	0.28 ± 0.15	0.37 ± 0.25	0.09 ± 0.27	0.87	0.279
Melatonin	7.74 ± 9.86	4.55 ± 3.03	−3.32 ± 10.53	−0.63	0.530
DHEA-s	5.09 ± 3.92	3.67 ± 2.59	−1.42 ± 2.07	−2.06	0.039
Ratio of DHEA-s/Cortisol	20.92 ± 17.55	14.91 ± 12.72	−6.01 ± 20.06	−1.08	0.279

DHEA-s: dehydroepiandrosterone sulfate.

**Table 5 ijerph-19-03130-t005:** The results of satisfaction survey after forest healing program.

	Mean ± SD
Satisfaction	
Overall	4.23 ± 0.599
Outdoor activities	4.46 ± 0.877
Dry foot bath	4.54 ± 0.660
Drinking tea-conversation	4.46 ± 0.776
Woodworking	4.46 ± 0.519
Warm-up exercise	4.46 ± 0.519
Walking	4.46 ± 0.660
Singing bowl meditation	4.15 ± 0.801
Intention to Re-Participation	
Outdoor activities	4.23 ± 0.927
Dry foot bath	4.46 ± 0.660
Drinking tea conversation	4.38 ± 0.768
Woodworking	4.00 ± 0.070
Warm-up exercise	4.00 ± 0.707
Walking	4.31 ± 0.751
Singing bowl meditation	3.92 ± 0.954
Intention to Recommend to Others	
Outdoor activities	4.31 ± 0.947
Dry foot bath	4.38 ± 0.650
Drinking tea conversation	4.38 ± 0.650
Woodworking	4.15 ± 0.801
Warm-up exercise	4.23 ± 0.725
Walking	4.38 ± 0.768
Singing bowl meditation	4.15 ± 0.801

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
