# Peer review of "Positive Effects on Emotional Stress and Sleep Quality of Forest Healing Program for Exhausted Medical Workers during the COVID-19 Outbreak"

_ijerph, 2022, doi:10.3390/ijerph19053130_

Round 1

Reviewer 1 Report

This manuscript represents a study examining the benefits of forest healing for medical worker burnout during the pandemic. 

The sample size is very small and the results are interesting.  Some additions could improve the paper. 

  1. Inclusion of norms for measures of all psychological and physiological variables used in the study.
  2. Description of the shift workers; were they evening shift or night shift.  Were the shift workers the ones who did not show any melatonin?
  3. Description of regular daylight schedule workers.
  4. How different was bedtime at the camp from the participants' at home schedule?  5. Are the changes in psychological and physiological measures clinically relevant? 

Author Response

Point 1: Inclusion of norms for measures of all psychological and physiological variables used in the study.

Response 1: The norm of measurements used in this study was presented.

Point 2: Description of the shift workers; were they evening shift or night shift.  Were the shift workers the ones who did not show any melatonin?

Response 2:  As suggested in this study, shift workers are a common medical work type in Korea, which means three shifts. Two clinical pathologists worked in three shifts. Shift workers also showed melatonin.

Point 3: Description of regular daylight schedule workers.

Response 3: The schedule of the day worker was 9-6. The schedule of the day worker was 9-6.2 shift workers reported that there was no significant difference from day work. Other weekly workers also did not show much difference from the average bedtime of the previous seven weeks because the camp period was only one night.

Point 4: How different was bedtime at the camp from the participants' at home schedule?

Response 4: It did not give much meaning because it was a change in sleep time of only one day. However, reviewing the questionnaire again, two-shift workers reported similar sleeping hours to daytime working hours, and 12 other participants also reported no significant change in average sleeping time during the week of forest healing participation. However, this paper did not objectively mention it. We are planning a follow-up study, and in the follow-up study, we will further review the part you suggested and study it with a more significant number of participants if there is a difference even though it is one day of silk and forest healing. Thank you again for the excellent opinion.

Point 4: Are the changes in psychological and physiological measures clinically relevant? 

Response 4: In terms of improving sleep quality, the final goal of this study, the results of this pilot study, are clinically related. Sleep quality is a subjective appeal, so it is clinically more important than any objective indicator (e.g., sleep latency, absolute sleep time).

Reviewer 2 Report

This is a potentially interesting work on the effects of forest therapy on wellness of medical workers subjected to emotional exahustion during the COVID-19 era. Clues of benefits emerge from both questionnaire surveys and analytical measures of cortisol and other stress-related indicators pre and poster forest therapy. 

It could be accepted in my opinion if authors convincingly respond to the following questions/requests of revisions.

-line 30 '
 a decrease is work performance ' probably should be '
 a decrease in work performance '

-change 'burnout' everywhere (Title and text of the manuscript' with 'exhaustion'. For example: emotional exhaustion, etc

-ine 63: provide more details on the conditions under which the program took place, the month and year, the season, the number of light hours/day, the average diurnal temperature outside 

-provide a reference after 'salivary cortisol, melatonin, dehydroepiandrosterone sulfate (DHEA-S)'. line 65

-line 79: provide country information for Hoengseong 

-section 2.3: provide references after 'Participants conducted assessments of emotion, sleep state, somatization using the Epworth sleepiness scale (ESS), the Stanford sleepiness scale (SSS) (to evaluate the day-time somnolence) '

-line 99: check 'bed. , '

-line 54-56: more literature examples on the importance of forest healing, and forest bathing together with sport on human health should be givenin the revised Introduction. Breathing volatile compounds from plants can have effects on human health and thus 'forest bathing' is considered a beneficial practice against COVID-19-related disorders. 

Even indoor plants in houses and offices can help people to bolstering their immune system.  Comment on the above points and cite at least the two works indicated by DOI above.

-Tab 4 DHEA-S: considering error bars I do not see significant difference in pre and post. Same for PSS in Tab 3

-Line 208: This is a small-sized study. Provide literature references of works published on the basis of similar small numbers of participants

-Line 217: consider to give Limitations and Strengths of the study in a separate section

Author Response

Response to Reviewer 2 Comments

Point 1: line 30 '
 a decrease is work performance ' probably should be ' a decrease in work performance

Response 1: line 30

Thank you for your careful review. We have modified "decrease in work performance".

Point 2: -change 'burnout' everywhere (Title and text of the manuscript' with 'exhaustion'. For example: emotional exhaustion, etc

Response 2: After discussing with the researchers, we changed 'burnout' to 'exhaustion', reflecting the reviewer's opinion.

Point 3: ine 63: provide more details on the conditions under which the program took place, the month and year, the season, the number of light hours/day, the average diurnal temperature outside 

Response 3: Line 86 provided information on the area to which the forest healing center where the research program was conducted belongs.

Point 4: provide a reference after 'salivary cortisol, melatonin, dehydroepiandrosterone sulfate (DHEA-S)'. line 65

Response 4: According to the reviewer's opinion, references were presented after each biomarker.

Point 5: line 79: provide country information for Hoengseong 

Response 5: Line 86 provided information on the area to which the forest healing center where the research program was conducted belongs. And line 85 provided information of the province information for Hoensung.

Point 6: section 2.3: provide references after 'Participants conducted assessments of emotion, sleep state, somatization using the Epworth sleepiness scale (ESS), the Stanford sleepiness scale (SSS) (to evaluate the day-time somnolence)

Response 6: References were presented at the end of the sentence. However, according to the reviewer's opinion, a renewal was presented after each biomarker.

Point 7: line 99: check 'bed.

Response 7: We checked the content. We erased the comma.

Point 8: line 54-56: more literature examples on the importance of forest healing, and forest bathing together with sport on human health should be givenin the revised Introduction. Breathing volatile compounds from plants can have effects on human health and thus 'forest bathing' is considered a beneficial practice against COVID-19-related disorders. 

Response 8: Introduction suggested that forest bathing has a positive effect on human physical and mental health, and reference was also suggested.

Point 9: Even indoor plants in houses and offices can help people to bolstering their immune system.  Comment on the above points and cite at least the two works indicated by DOI above.

Response 9: As the reviewer said, indoor plants can also help strengthen the immune system. However, this paper's focus is not on horticultural therapy at home or in the office but on the health status of subjects before and after conducting a healing program in the forest space so that it will be reflected in future horticultural treatment studies. Thank you.

Point 10: Tab 4 DHEA-S: considering error bars I do not see significant difference in pre and post. Same for PSS in Tab 3

Response 10: As shown in the figure, there were outliers, so the error bar of DHEA-S seemed to have a significant difference after the pre-post, but it was statistically significant, as shown in Table 4.

Point 11: Line 208: This is a small-sized study. Provide literature references of works published on the basis of similar small numbers of participants

Response 10: As the reviewer said, we looked for papers published based on a small number of similar participants. Unfortunately, we could not find papers published similarly.

Point 12: Line 217: consider to give Limitations and Strengths of the study in a separate section

Response 12 : Lines 238 to 243 suggest the limitations of the areas suggested by the reviewer and future research directions.

“Sixth, the sample size was small. As this study is a pilot study, the statistical significance of psychological factors and sleep could not be confirmed due to the small size of participants. A large-scale study is thus required in order to analyze the detailed effects of stress and sleep quality improvement by conducting research over various periods and seasons, using a more varied sample of medical workers.

Round 2

Reviewer 1 Report

The authors did not respond adequately to my previous comments; no new statistics were presented to show clinical relevance, differences between sleep schedules, etc.  

Author Response

Point 1: The authors did not respond adequately to my previous comments; no new statistics were presented to show clinical relevance, differences between sleep schedules, etc.

Response 1: We are sorry that we could not properly reflect your review. So, we analyzed the sleep onset time and sleep duration before and after the forest healing program. As a result of the analysis, there was no statistically significant difference in sleep onset time before and after the forest healing program. Still, there was a statistically significant difference in sleep duration. You can see the analyzed results in the Table 3 and results.
